# Genetic Evaluation of Resilience Indicators in Holstein Cows

**DOI:** 10.3390/ani15050667

**Published:** 2025-02-25

**Authors:** Eva Kašná, Ludmila Zavadilová, Jan Vařeka

**Affiliations:** 1Institute of Animal Science, 10400 Prague, Czech Republic; kasna.eva@vuzv.cz; 2Department of Genetics and Breeding, Faculty of Agrobiology, Food and Natural Resources, Czech University of Life Sciences Prague, 16500 Prague, Czech Republic; vareka@af.czu.cz

**Keywords:** dairy cattle, genomic selection, resilience, single-step genomic prediction

## Abstract

Climate change jeopardises the sustainability of agricultural systems. Therefore, a response to these changes is needed. The sustainability of dairy cattle is closely related to the concept of resilience. Resilience in livestock is defined as the ability to resist or be only minimally affected by short-term disturbances and to return to a healthy state as quickly as possible after these disturbances end. A trait that is frequently measured and that shows a response to disturbances is milk yield. Resilience indicators can be expressed as a variance of milk yield or as its deviation from optimal production (residuals). Resilience indicators are characterised by low-to-intermediate heritability. A strong genetic correlation exists between daily milk yield variance and the variance of its residuals. Better resilience was associated with lower milk yield but higher milk protein and milk fat contents. Resilience indicators were genetically related to production, reproduction, and functional traits to different extents and could be included in the selection index. In the future, breeding should focus more on resilience traits, as they are closely related to the economic self-sufficiency of farms.

## 1. Introduction

Dairy cattle farming faces two major challenges: requirements to reduce its climate impact and meeting society’s growing demands for the production of healthy, nutritious food [1]. At the same time, climate change significantly affects the sustainability of the dairy sector. Increasing temperatures and prolonged droughts directly affect fodder/pasture quality and quantity, the availability of water, heat stress, and pest and parasite pressure [2]. The physiological responses induced by heat stress significantly affect production sustainability, as well as animal reproduction, health, and welfare [3]. The severity of these impacts is closely related to resilience, which is defined as the capacity of animals to cope with short-term perturbations in their environment and to return rapidly to their pre-challenge status [4,5]. The correct and timely identification of resilient animals will allow the optimization of breeding, treatment, and culling decisions [6]. The inclusion of resilience in the selection index will result in less labour-demanding and easier-to-manage livestock [5]. However, many modern cattle breeding goals have negative genetic gains for resilience, and adding resilience to the selection index in a simulation study led to a 12–76% loss in the aggregated genotype, mainly due to the reduction in genetic gain in milk production [7].

Resilience is not directly measurable because of its complexity [4]. Therefore, various indicators have been proposed to quantify and compare its levels among individuals. The most common resilience indicators in dairy cattle are those derived from fluctuations in daily milk yield (DMY). Records of milk yields are readily available because most farmers use automatic measurement systems or automatic milking systems (AMSs). As milk yields reflect the physiological status of cows, individuals experiencing a disturbance produce less than expected, and resilience might be measured based on the variance of milk yield [8] or the variability of its deviation from either mean performance [9] or predicted performance [10,11,12].

It is not certain whether resilience indicators determined on the basis of the whole lactation period will also indicate resilience during whole lactation [13]. Resilience indicators derived from specific lactation periods can be more specific, which is beneficial because susceptibility to disturbances is not static, but rather changes throughout lactation. This variability in disturbance susceptibility highlights the need to research specific parts of lactation. In this study, we concentrate on the period between 50 and 150 days in milk (DIM) when great demands are placed on the cow due to the requirements for milk production and, simultaneously, for getting pregnant again.

Several studies have shown that the natural log-transformed variance of daily deviations from fitted lactation curves (LnVar) is a promising general resilience indicator [9,10,11,12,13]. LnVar indicates the impact of the disturbance, and its heritability estimates range between 0.13 and 0.25 [10,11,12]. LnVar is also strongly genetically correlated with health, longevity, fertility, metabolic, and production traits, and a lower variance is related to better udder health, better hoof health, better longevity, better fertility, higher body condition scores, less ketosis, and lower milk yields [10]. Another aspect of resilience is expressed by the lag-1 autocorrelation of daily deviations from fitted lactation curves (r-auto), which indicates the length of recovery time. Its heritability is estimated to range between 0.01 and 0.10, and genetic correlations with other traits are weaker compared with LnVar [10,11]. The described resilience indicators are associated with short-term disturbances.

The objectives of this study are to estimate genetic parameters and predict genomic breeding values (GEBVs) for resilience indicators in mid-lactation Holstein dairy cows, to estimate the genetic correlations among these indicators, and to approximate genetic correlations with traits included in the Czech Holstein selection index (SIH) [14].

## 2. Materials and Methods

DMY data were recorded at 10 commercial dairy cattle farms between 2022 and 2024. Six farms were equipped with AMSs; three with conventional milking parlours; and one with an automatic rotary milking parlour. The original dataset contained 2,067,538 milk yield records of 7041 cows from 14,891 lactations. Cows with at least 75% Holstein breed adherence were used in the analysis. Records from the lactation period of 50–150 DIM were extracted. To avoid additional variations in DMY derived from irregular AMS visits, data were adjusted to daily 24 h yields, as described in [15]. DMY records were removed when they were <2 kg or >90 kg. Contemporary groups of cows were defined by concatenating calving herd, year, and season. The evaluated dataset contained 329,589 DMY records from 3347 lactations of 3080 Holstein cows. The descriptive statistics of the data are shown in Table 1.

A random regression function with a fourth-degree Legendre polynomial was used to predict the lactation curve. The Proc Reg procedure (SAS/STAT software, v9.4; SAS Institute, Inc., Cary, NC, USA) was used to predict the individual daily performance of each dairy cow. After obtaining the fitted lactation curve, daily deviations were calculated as $DMY_{t}-\hat{\;DMY_{t}}$, where $\hat{\;DMY_{t}}$ is the expected DMY at *t* DMI. The resilience indicators were calculated as the log-transformed variance of DMY (Var), log-transformed variance of daily deviations (LnVar), lag-1 autocorrelation of daily deviations (r-auto), and skewness (skew) of daily deviations from the fitted individual lactation curves. These indicators represent the animal phenotypes used for genetic analysis in this study, i.e., the estimation of genetic parameters and prediction of GEBVs. As predicted, lower fluctuation and thus lower values of all four indicators indicated more resilient cows.

The recorded systematic effects (herd, year and season of calving, parity, age at calving) were evaluated using the generalised linear model method Proc Glm (SAS/STAT software, v9.4; SAS Institute, Inc., Cary, NC, USA). All of these effects were significant and were included in the linear mixed model equation for genetic evaluation, as follows:*y_ijklm_* = *µ* + *P_i_* × *age_j_* + *HYS_k_* + *a_l_* + *pe_m_* + *e_ijklm_*,(1)
where

*y_ijklm_* is the evaluated indicator, i.e., Var, LnVar, r-auto, or skew;*μ* is population mean;*P_i_ × age_j_* is the fixed effect of parity *i* (five levels) combined with the age at calving *j* (three classes according to quantiles 10, 90, and the rest), with fifteen levels;*HYS_k_* is the fixed effect of herd–year–season *k* (10 herds, 3 years, and 3 seasons—38 levels in total);*pe_m_* is the random effect of the permanent environment of the cow *m* (3080 levels);*a_l_* is the random effect of individual *l*, connected with 3-generation pedigree (31,799 levels);*e_ijklm_* is the random residual.

The random effect of animals was supplemented with SNP genotypes (Illumina BovineSNP50 BeadChip, San Diego, CA, USA) for 1764 cows with DMY records. Other genotypes for pedigree animals were retrieved from a genomic matrix used for a national routine genetic evaluation (Plemdat, Hradistko, Czech Republic). A total of 9845 genotyped animals and 36,839 effective SNPs were included in the estimation.

The variance components of the resilience indicators were estimated using model Equation (1), utilising a multi-trait mixed linear animal model to assess their relationships. The estimation used the average information REML method implemented in REMLF90 software [16,17]. GEBVs were predicted using the single-step genomic prediction method ssGBLUP and the programme BLUP90IOD [17]. The reliability of the GEBVs was approximated using the ACCF90GS programme, 2024 version [17].

GEBVs were converted to relative breeding values (RBV) with mean = 100 and SD = 12 for base bulls born in 2010:*RBV* = *(GEBV − mean of the genetic base/SD)* × 12(2)

As higher values are generally perceived as desirable, we reversed the sign in RBV for Var, LnVar, and r-auto so that higher values are associated with better resilience.

The genetic relationships between resilience indicators and other traits in the SIH were approximated using Pearson correlation coefficients between GEBVs of sires, with GEBV accuracy >0.35 [12]. For other traits, we used sire GEBVs from the official run of December 2024 as input (Czech Moravian Breeding Corporation, Hradistko, Czech Republic). As correlations underestimate actual genetic correlations, unless GEBV accuracies are close to 1, the results show only general indications.

## 3. Results

### 3.1. Phenotypic Characteristics of Resilience Indicators

The descriptive statistics of the resilience indicators are shown in Table 2. Autocorrelation values were used regardless of the sign (absolute values), which better describes the fact that better resilience is associated with low values, i.e., close to zero.

The normal distributions of the resilience indicators are shown in Appendix A.

Figure 1 shows the average course of the monitored part of lactation. The average DMY was lower in primiparous than in older cows. The lactation curve of primiparous cows was flat, while the daily milk yield of older cows decreased from the peak yield by an average of 6 kg of milk over 100 days.

### 3.2. Genetic Parameters

The variance components of the resilience indicators estimated using single-trait animal models (Equation (1)) are shown in Table 3. The results showed the highest additive genetic variance in both DMY variance-based indicators. Both indicators also had comparable phenotypic variance, leading to similar estimates of heritability (Table 4). In contrast, very low additive genetic variance was found for r-auto.

The associations among resilience indicators were estimated using a multi-trait linear animal model (Table 4). Heritability estimates were generally low, and they ranged from 0.02 (skew) to 0.13 (LnVar). The strongest phenotypic and additive genetic correlations were found between both variance-based indicators. The genetic associations between both variances and r-auto were moderate and positive, and with skew they were negative. The genetic correlation between r-auto and skew was also strong but negative.

### 3.3. GEBV and Correlations with Production, Exterior, Reproduction, and Health Traits

Appendix A show the basic characteristics of the GEBVs and the accuracy of the GEBVs. The tables are divided by animal sex, and special groups represent genotyped animals and cows with phenotypes for resilience indicators.

Figure 2 shows the average accuracy of the GEBVs for resilience indicators and particular groups of animals. The accuracy was, on average, the highest for Var and LnVar and for genotyped animals, especially for cows with a phenotype for resilience. The lowest average accuracy was that for skew. The highest average accuracy values were approximately 0.25 and 0.30, and the lowest was slightly over 0.10. Higher accuracy values were determined for sires with daughters. The number of daughters of sires with accuracy equal to and over 0.50 was from 25 (LnVar) to 42 (skew).

The correlations between RBVs for resilience indicators and yield traits (Figure 3) were negative, indicating a higher variability in milk yield and worse resilience in high-yielding cows. On the contrary, the associations between resilience and milk fat and protein contents were positive. A negative correlation with production traits was also reflected in the negative associations between RBVs for most resilience indicators and the SIH. The SIH combines 0.49 production (0.28 kg protein, 0.135 kg fat, 0.055 protein content, and 0.02 fat content), 0.24 exterior (0.13 udder and 0.11 feet and legs), and 0.27 functional traits (0.15 fertility, 0.07 somatic cells, and 0.05 longevity).

Pearson correlation coefficients with exterior traits, which determine the functional type of dairy cows and are used in the calculation of the SIH, are presented in Figure 4. The correlations between resilience indicators and feet- and leg-type traits were mostly positive, with the strongest correlations (~0.39) found between skew and foot and leg scores and between skew and locomotion. Correlations with udder-type traits were weak, with two exceptions (Var—central ligament; r-auto—rear teat placement) not exceeding 0.20.

The correlations between resilience indicators and daughters’ fertility (heifers, cows, and both groups together) were weak to medium. Better fertility was related to better resilience indicated by LnVar and Var, while the correlations with skew and r-auto were opposite negative (Figure 5). The ability to be born easily (direct calving ease) was related to better resilience, while the correlations with maternal calving ease (ability to give birth easily) were negligible.

The correlations between longevity index and RBVs for the resilience indicators were unfavourable (Figure 6). This result was affected by the composition of the index, which combines longevity with the fertility of cows, body depth, udder depth, foot and leg score, and somatic cell count. Unfortunately, breeding values for longevity alone are not published and they were not available for analysis. Better resilience was associated with better body condition score; better resistance to clinical mastitis; better resistance to claw horn lesions (Var and LnVar); and with a better health index, consisting of resistance to clinical mastitis and resistance to claw diseases/disorders.

## 4. Discussion

We analysed four resilience indicators related to DMY fluctuations. Previous studies [11,13] have shown that calculating variability traits from the entire lactation period might give poor results due to missing data and poor fitted lactation curves during the early and late lactations. We therefore focused on milk yields in 50–150 DIM, when the cow is challenged by the peak of lactation and the exposure to general stress and susceptibility to risk factors might increase [18]. The estimates of genetic correlations across lactation stages also indicated that resilience may differ genetically throughout lactation [11,13]. Early-lactation (transition) cows prioritise milk synthesis, suffer more with immune suppression, and have increased disease susceptibility [19], while mid-lactation cows are more liable to heat stress [20] and hoof diseases/disorders [18,21].

We considered resilience indicators in different lactations as repeated traits and used a repeatability model for evaluation. However, previous studies were mostly based on records of primiparous cows only [8,10], or they treated resilience in different lactations as different traits [12,13]. The small size of the analysed dataset due to the short observation period and the limited number of farms involved was the reason for the use of the repeatability model. If a larger number of lactations was achieved for different parities, the results of the resilience indicators in the first and subsequent lactations would be worth comparing.

### 4.1. Genetic Parameters of Resilience Indicators

The coefficients of heritability of the indicators in our study were comparable to those reported in [11] for the peak lactation period. As reported in [13], resilience indicators genetically differ across lactation, and the estimates of heritability are usually higher based on full lactation and lower in different lactation periods. The strongest genetic correlation was found between DIM 111–210 and DIM 211–340, and the association of both periods with DIM 11–110 was weak. These correlations may reflect the different genetic backgrounds of resilience in transition periods compared with the rest of lactation.

The genetic correlations among the analysed resilience indicators (Var, LnVar, r-auto, skew) suggest that they account for different aspects of resilience, such as low vulnerability to disturbance or quick recovery [5,10]. In general, the association between Var and LnVar was the strongest, as they both indicated the impact of disturbances [5,10]. The strong negative genetic association between skew and r-auto, indicating slow recovery from negative deviations caused by disturbances, was previously reported in [10] and in [11] for the peak lactation period (DIM 60-90). Our data also confirmed the positive genetic correlation between LnVar and r-auto, expressing slower recovery from deeper-impact disturbances [10,11,12]. Similar to [11], we found a positive though weaker association between skew and LnVar, contrary to [10], which observed a moderate positive correlation between these two indicators.

### 4.2. Corelations of Resilience Indicators with Other Traits

The negative correlations between yield traits and indicators showed that cows with high production had more variable DMY. This corresponds to a more pronounced decrease in milk yield after lactation peaks in older cows compared with the relatively flat lactation curve of heifers (Figure 1). A strong genetic correlation between average milk yield, Var, and LnVar, was previously reported in [10], which also found a moderate genetic correlation between LnVar and official milk yield produced at 305 days of lactation. The genetic correlations between DMY and r-auto ranged from low to moderate [10,11,12] and varied depending on the lactation period and the parity of the cow. The strongest association between these traits was observed during DIM 10–110 [13], which was close to the lactation period analysed in our study. The genetic correlations between milk yield and skew reported in [10] were moderate, and their direction depended on the method of fitting of the lactation curve. Contrary to our study, a positive moderate genetic correlation between skew and milk yield at 305 days was estimated in [11], suggesting mainly positive responses to environmental improvements [5].

Most studies assume that better resilience, as indicated by fewer fluctuations in DMY, is associated with better fertility [6,7]. The association between Var, calving interval, and the interval from first to last insemination was found to be favourable [8]. Another study [10] reported that lower Var, LnVar, and r-auto were genetically correlated with better fertility, and that correlations with skew were negligible or weak. In addition, Ref. [11] reported positive correlations between LnVar and reproductive intervals (from calving to insemination and from first to last insemination), a negative relationship between r-auto and the interval from first to last insemination, and a moderate positive correlation between skew and interval from calving to the first insemination. On the contrary, Ref. [12] estimated a positive correlation between LnVar and daughter pregnancy rate and a negligible correlation between r-auto and daughter pregnancy rate in the first and second lactations. In our study, we found a favourable association of daughter fertility with both variance-derived resilience indicators (Var, LnVar), while the correlations with r-auto and skew were negative. The unfavourable relations may reflect the resource competition and the balance between production and reproduction in the observed period after peak yields. High-producing cows with persistent lactation may suffer more from ovulation disorders [22] and show a weak expression of oestrus behaviour [23], which leads to extended calving intervals and reduced fertility.

To assess the biological relevance of these indicators, their association with health traits and longevity is often used. Resilient animals should be healthy, avoiding early culling by coping well with a farm’s conditions [6]. In our study, most indicators showed a favourable correlation with the traits included in the health index, particularly with resistance to clinical mastitis and to claw lesions. The association with claw lesion resistance is consistent with the favourable correlations of resilience indicators with foot and leg traits (Figure 4). This association might also reflect the higher liability of cows to claw diseases, such as sole haemorrhage or sole ulcers, in a given lactation stage, e.g., DIM50–150 [18,21]. Similarly, Ref. [13] reported favourable genetic correlations between LnVar, Var, and r-auto and hoof and udder health, positive correlations of skew with udder health traits, and negligible correlations of skew with hoof health.

Positive correlations of resilience indicators with body condition score might reflect that cows with enough body reserves could use them to better respond to environmental disturbances [7].

In contrast to our results, most studies [8,10,12] found a favourable genetic relationship between resilience indicators and longevity. However, Ref. [11] reported a high positive genetic correlation between LnSD, 305 days of milk yield, and longevity, indicating that high-yielding cows are less resilient, but avoid active culling associated with poor production and tend to have a longer productive life.

Owing to the high correlation values between the indicators and traits analysed, Ref. [13] recommends both LnVar and r-auto based on overall DMY during lactation as suitable resilience indicators. However, they also found high correlation values for r-auto when analysing the lactation period of 10–100 DIM, which is close to the lactation period that we studied. This could support our theory that this part of lactation is sufficient and suitable for evaluating resilience. Our research yielded generally low correlation values due to a different calculation method compared with that reported in [13], although we found some high values for r-auto and skew. Therefore, we assume that both r-auto and skew can serve as suitable resilience indicators in cattle, although skew is sensitive to extreme milk yields and needs a strict quality control and fitting procedure [11]. These indicators could also be used in genomic selection to increase the general resilience of dairy cows. Including resilience in breeding goals would help to reduce costs related to labour and health treatments, mainly on farms with large herds [5]. However, their use must be cautious in view of the unfavourable genetic correlations with production traits. As shown in [7], increasing the economic weight on LnVar in terms of breeding goals led to improving true resilience, but also to a loss in genetic gain, mainly due to a reduction in milk production.

## 5. Conclusions

We evaluated four resilience indicators derived from fluctuations in DMY in mid-lactation Holstein cows. Although cow performance fluctuated relatively less over the studied period compared to the whole lactation, the selected indicators enabled the evaluation of the ability of cows to cope with environmental perturbations. The strong genetic correlation between variance-based indicators showed that they contained overlapping information on resilience. A lower correlation with r-auto suggests that a resilience sub-index would be beneficial for use in selection. General resilience indicators would complement the existing health index well, which already includes resistance to clinical mastitis and hoof disorders. Distinguishing between highly resilient cows and cows with low resilience can be used for herd management or selection. However, genetically selecting a highly resilient population will lead to the selection of resistant dairy cows with stable but low performance. Therefore, it is necessary to consider the breeding goals of the breed.

## Figures and Tables

**Figure 1 animals-15-00667-f001:**
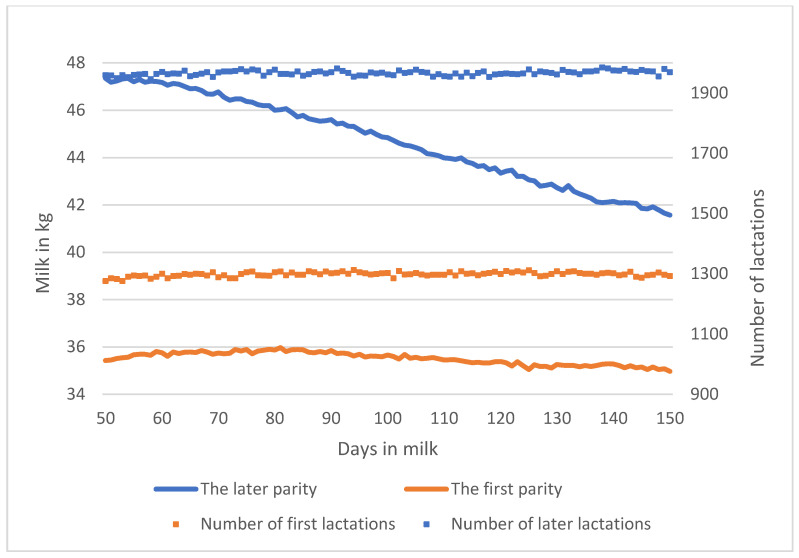
Average daily milk yield and number of animals in the first and later parities.

**Figure 2 animals-15-00667-f002:**
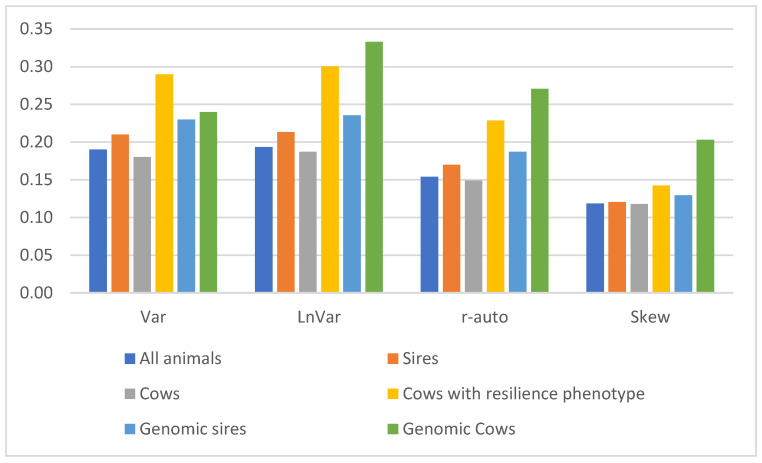
Average accuracies of GEBVs for resilience indicators in particular groups of animals. Resilience indicators are log-transformed variance of daily milk yields (Var), log-transformed variance of deviations (LnVar), lag-1 autocorrelation of deviations (r-auto), and skewness of deviations (skew).

**Figure 3 animals-15-00667-f003:**
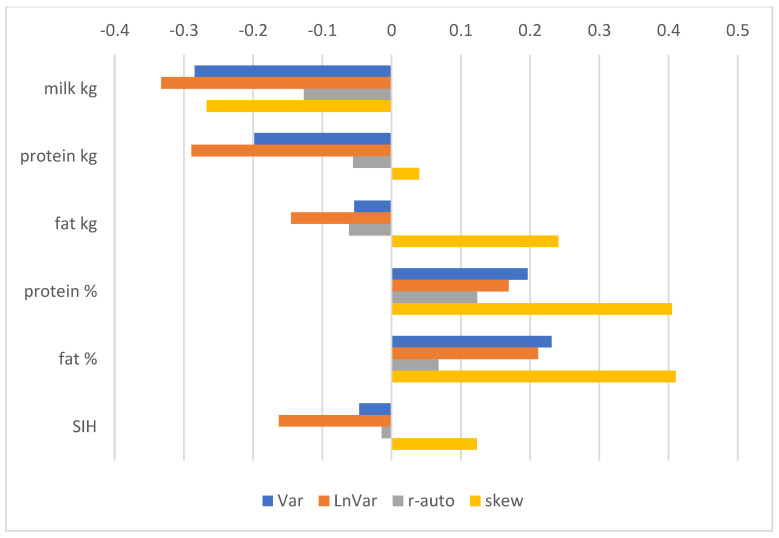
Pearson correlation coefficients between RBVs of sires considering resilience indicators and RBVs considering milk production traits and the Czech Holstein Selection Index (SIH). Resilience indicators are log-transformed variance of daily milk yields (Var), log-transformed variance of deviations (LnVar), lag-1 autocorrelation of deviations (r-auto), and skewness of deviations (skew). Milk production traits include milk yield (kg), protein yield (kg), fat yield (kg), and protein and fat contents in milk (%). Higher RBVs are desirable; therefore, positive correlations are favourable.

**Figure 4 animals-15-00667-f004:**
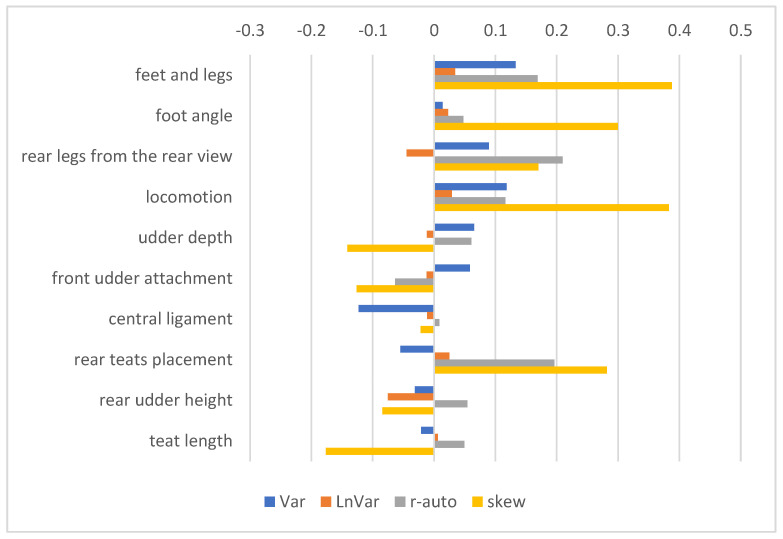
Pearson correlation coefficients between RBVs of sires considering resilience indicators and RBVs considering selected exterior traits. Resilience indicators were log-transformed variance of daily milk yields (Var), log-transformed variance of deviations (LnVar), lag-1 autocorrelation of deviations (r-auto), and skewness of deviations (skew). Exterior traits include four feet- and leg-type traits and six udder-type traits.

**Figure 5 animals-15-00667-f005:**
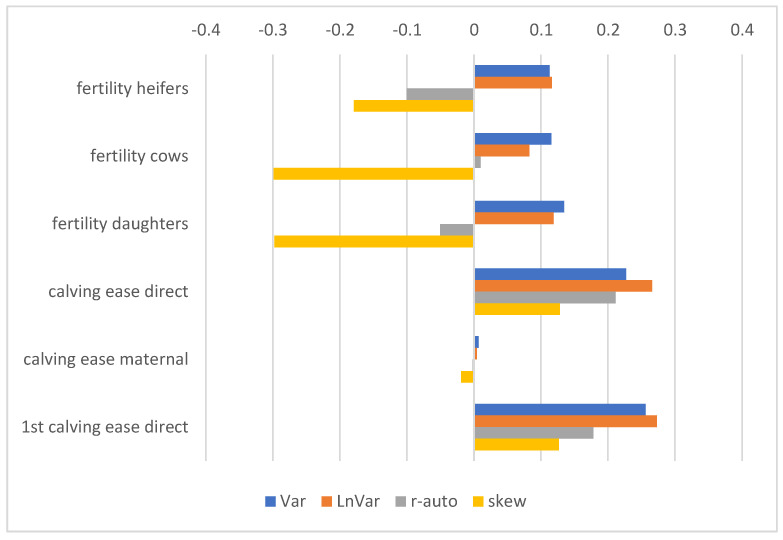
Pearson correlation coefficients between RBVs of sires considering resilience indicators and RBVs considering fertility traits. Resilience indicators are log-transformed variance of daily milk yields (Var), log-transformed variance of deviations (LnVar), lag-1 autocorrelation of deviations (r-auto), and skewness of deviations (skew). Fertility traits include daughter fertility recorded in heifers, cows, and in both groups (daughters) together, direct calving ease, maternal calving ease, and direct 1st calving ease. Higher RBVs are desirable; therefore, positive correlations are favourable.

**Figure 6 animals-15-00667-f006:**
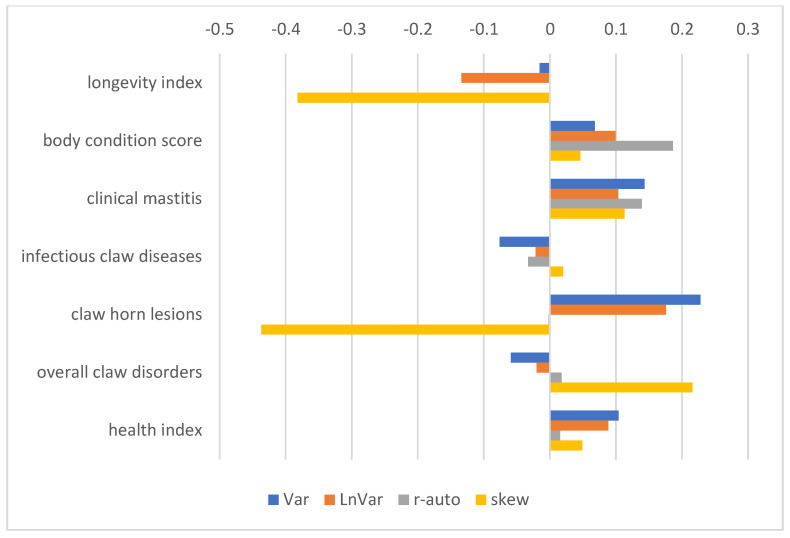
Pearson correlation coefficients between RBVs of sires considering resilience indicators, RBVs considering health traits, RBVs considering body condition score, and longevity and health indexes. Resilience indicators are log-transformed variance of daily milk yields (Var), log-transformed variance of deviations (LnVar), lag-1 autocorrelation of deviations (r-auto), and skewness of deviations (skew). Longevity index combines functional longevity with fertility of cows, body depth, udder depth, foot and leg score, and somatic cell count. Health index consists of resistance to clinical mastitis, infectious claw diseases, claw horn lesions, and overall claw disorders. Higher RBVs are desirable; therefore, positive correlations are favourable.

**Table 1 animals-15-00667-t001:** Number of records and descriptive statistics of evaluated data.

	N	Minimum	Maximum	Mean	SD	Coefficient of Variation
DMY (kg)	331,589	5.05	80.90	41.04	74.59	21.05
DMY conventional (kg)	85,722	5.10	80.90	41.45	63.41	19.21
DMY-AMS (kg)	169,350	5.10	79.25	40.61	91.02	17.02
DMY robotic parlour (kg)	76,517	5.05	78.88	41.52	49.91	23.49
DIM (days)	331,589	50	150	100.03	849.60	29.14

DMY—daily milk yield in kg; DMY conventional—recorded via conventional milking system; DMY-AMS—recorded via automated milking system; DMY robotic parlour—DMY recorded via robotic rotary parlour milking; DIM—days in milk; N—number of records.

**Table 2 animals-15-00667-t002:** Descriptive statistics of analysed resilience indicators.

Indicator	No	Minimum	Maximum	Mean	Variance	Coefficient of Variation
Var	3347	–0.46	5.69	2.41	0.67	34.01
LnVar	3347	–1.12	5.06	1.88	0.55	39.62
r-auto	3347	0.00	0.90	0.32	0.04	62.83
skew	3347	–5.73	5.63	–0.81	1.16	–132.91

Var—log-transformed variance of daily milk yields; LnVar—log-transformed variance of deviations; r-auto—lag-1 autocorrelation of deviations; skew—skewness of deviations.

**Table 3 animals-15-00667-t003:** Estimated variance components of resilience indicators (SD in parentheses).

Variance Component	Var	LnVar	r-Auto	Skew
$\sigma_{a}^{2}$	0.057 (0.015)	0.051 (0.014)	0.002 (0.0009)	0.024 (0.012)
$\sigma_{pe}^{2}$	0.019 (0.020)	0.036 (0.021)	0.004 (0.0021)	0.024 (0.042)
$\sigma_{e}^{2}$	0.404 (0.023)	0.317 (0.022)	0.027 (0.0022)	1.005 (0.050)
$\sigma_{p}^{2}$	0.480	0.404	0.034	1.053

Var—log-transformed variance of daily milk yields; LnVar—log-transformed variance of deviations; r-auto—lag-1 autocorrelation of deviations; skew—skewness of deviations. Variance components included variances due to the direct additive genetic effect ($\sigma_{a}^{2}$), permanent environment of the cow ($\sigma_{pe}^{2}$), residual error variance ($\sigma_{e}^{2}$) and total phenotypic variance ($\sigma_{p}^{2}$). SD—posterior standard deviation by GIBBSF90.

**Table 4 animals-15-00667-t004:** Estimated heritability (on the diagonal), genetic (above the diagonal), and phenotypic correlations (below the diagonal) among resilience indicators.

Trait	Var	LnVar	r-Auto	Skew
Var	12	87	56	−43
LnVar	92	13	26	−20
r-auto	17	13	7	−73
skew	9	11	−58	2

Var—log-transformed variance of daily milk yields; LnVar—log-transformed variance of deviations; r-auto—lag-1 autocorrelation of deviations; skew—skewness of deviations.

## Data Availability

The raw data supporting the conclusions of this article will be made available by the authors on request.

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
