# Peer review of "Genetic Evaluation of Resilience Indicators in Holstein Cows"

_animals, 2025, doi:10.3390/ani15050667_

Round 1

Reviewer 1 Report

Comments and Suggestions for Authors

The paper deals with the genetics of resilience in cattle. It was well made, and brings useful new knowledge. The study of genetic regulation of health is crucial task for geneticists and breeders. I have only comment concerning recommendation for practical use, try to flesh out it a bit in Conclusions and Abstract.

Formal error, in Introduction, explain LnVar and r-auto, rr. 70, 75.

Author Response

Dear Reviewer,

Thank you for reviewing our manuscript and providing corrections and comments that will help improve its quality. We have accepted almost all suggested corrections, and the text has been revised accordingly.

Comments and Suggestions for Authors

The paper deals with the genetics of resilience in cattle. It was well made, and brings useful new knowledge. The study of genetic regulation of health is crucial task for geneticists and breeders. I have only comment concerning recommendation for practical use, try to flesh out it a bit in Conclusions and Abstract.

Suggestion: : recommendation for practical use

Answer: The distinction between highly resilient and low-resilient cows can be used for herd management or selection. However, selection for a highly resilient population will lead to the selection of resistant dairy cows with stable performance, but at the same time with lower performance. Therefore, it is necessary to consider the breeding goal of the breed.

That sentence was included into conclusion (L373-377) and highlight in yellow. The abstract was not changed because of the 200-word limit, which has already been exceeded.

Suggestion: Formal error, in Introduction, explain LnVar and r-auto, rr. 70, 75.

Answer: Both LnVar and r-auto, rr. 72 and 79, were explained in the text and highlighted in yellow.

Reviewer 2 Report

Comments and Suggestions for Authors

Throughout is would seem important to distingish between the effects of climate (long term or over years) and weather (short term or day to day). The article deals primarily with the latter. 

In addition, the reader without know what caused changes in the resilience indicators over the course of the 100-day measurement period.  The study would be much improved if the mechanisms causing changes in these indicator traits could be included.

The r-auto trait indicates recovery over a time horizon of 1 day. What if it takes longer to recover from whatever disturbance? 

Correlations are technically in the range of -1 to 1. The figures present them after their being multiplied by 100. This might lead some readers to interpret them as percentages, which would be an incorrect interpretation. The same comment is relevent to the heritability estimates in Table 4.

Line 90: Herds were not considered as part of the contemporary group specification. This is highly unusual.

Line 96: Is a fourth-degree Legendre polynomial really needed to describe milk yield from day 50 to day 150 of lactation? This would seem doubtful.

Figure 1: omit "the" as it is not needed. Also insert space between "later" and "lactations". Spell out "number" rather than abbreviate it.

Table 3: It might be useful to present these variance components in relation to the trait means (ie, as coefficients of variation) in addition to their estimated values. This would help with understanding the maginute of the estimates.

Tables 3 and 4: There needs to be some indication of the variability in these parameter estimates.

Note that the correlations between EBV for traits are not equivalent to genetic correlations unless this accuracies of the EBV are 1., which is very much not the case in this study. The study would be improved if the genetic correlations with type, fertility, longevity and health traits were properly estimated.

Comments on the Quality of English Language

The article could be improved with some careful technical editing. There are numerous contract services to perform this function.

Author Response

Dear Reviewer,

Thank you for reviewing our manuscript and providing corrections and comments that will help improve its quality. We have accepted almost all suggested corrections, and the text has been revised accordingly.

Suggestion: Throughout is would seem important to distingish between the effects of climate (long term or over years) and weather (short term or day to day). The article deals primarily with the latter. 

Answer: Exactly, according to the definition of resilience (L46-49), we do not analyse the longterm effect but only random short-term effects. The sentence was added: The described resilience indicators are based on short-term disturbances (L 82, highlighted in blue).

Suggestion: In addition, the reader without know what caused changes in the resilience indicators over the course of the 100-day measurement period.  The study would be much improved if the mechanisms causing changes in these indicator traits could be included.

Answer: The sentence was added in lines 68-71, highlighted in blue: In this study, we concentrate on the period between the 50–150 days in milk (DIM) when great demands are placed on the cow due to the requirements for milk production and, simultaneously, for getting pregnant again.

Suggestion: The r-auto trait indicates recovery over a time horizon of 1 day. What if it takes longer to recover from whatever disturbance? 

Answer: R-auto wil be more pronounced.

Suggestion: Correlations are technically in the range of -1 to 1. The figures present them after their being multiplied by 100. This might lead some readers to interpret them as percentages, which would be an incorrect interpretation. The same comment is relevent to the heritability estimates in Table 4.

Answer: The values of correlations and heritabilities were corrected.

Suggestion: Line 90: Herds were not considered as part of the contemporary group specification. This is highly unusual.

Answer: Apologies for the error; the effect of the herd was added to the sentence and highlighted in blue (L95-96). We considered herd, year, and season of calving in the analysis, as stated in equation (3).

Suggestion: Line 96: Is a fourth-degree Legendre polynomial really needed to describe milk yield from day 50 to day 150 of lactation? This would seem doubtful.

Answer: We agree that it was redundant for the short part of lactation, and we could have used a third-degree polynomial.

Suggestion: Figure 1: omit "the" as it is not needed. Also insert space between "later" and "lactations". Spell out "number" rather than abbreviate it.

Answer: The description of Figure 1 was fixed.

Suggestion: Table 3: It might be useful to present these variance components in relation to the trait means (ie. as coefficients of variation) in addition to their estimated values. This would help with understanding the maginute of the estimates.

Answer: The Basic statistics of resilience indicators are in Table 2.

Suggestion: Tables 3 and 4: There needs to be some indication of the variability in these parameter estimates.

Answer: The analyses were done using Remlf90, therefore we no standard errors or variance span.

Suggestion: Note that the correlations between EBV for traits are not equivalent to genetic correlations unless this accuracies of the EBV are 1., which is very much not the case in this study. The study would be improved if the genetic correlations with type, fertility, longevity and health traits were properly estimated.

Answer: Thanks for pointing out that the Pearson correlations between RBVs are only approximation of actual genetic correlations. Unfortunately, the records of fertility, exterior and other traits that are necessary for estimation of genetic correlations were not at disposal for our study. Sentence was added (L151-153, blue): As correlations underestimate actual genetic correlations unless accuracies of GEBVs are close to 1, the results show only general indications.

Comments on the Quality of English Language

Suggestion: The article could be improved with some careful technical editing. There are numerous contract services to perform this function.

Answer: The manuscript was edited by Elsevier Language Editing Services before sending to Animals Editorial Office, as showed enclosed Certificate.

Reviewer 3 Report

Comments and Suggestions for Authors

In this study, the authors evaluated the resilience indicators in Holstein cows, and they used the multi-trait linear model equation to assess and correlate the indicators. I really appreciate the author’s efforts towards parameter selection to improve the adaptability of dairy cattle. Some sections are well-written and explained well, but I have some suggestions that are critical for the overall improvement of this manuscript.

The abstract section needs improvement in terms of explaining the hypothesis and objectives of the study, so readers can understand the significance of the study. The material and methods (M&M) and discussion sections also need major revisions. In terms of M&M, some equations like 1-4 are explained but I think they are already developed ones, especially equations 1-2, so if authors can reference them effectively then I do not think there is a need to explain them here. Equation 3, explaining the model should be included as explained but other equations if taken from previous studies then citing them will explain everything. The discussion section needs rewriting, although headings are good, there are too many headings for one section, and it should start generally and then move towards headings and limitations as well. The result sections are well written however, all tables and figure legends are not self-explanatory and need more details for the reader to understand them. Addressing these comments will contribute to the overall improvement of the manuscript. I also have some specific comments as mentioned below.

Specific comments:

1.      Lines 16-17: Improve the overall structure of the sentence.

2.      Lines 55, 69, and 81 need proper citations.

3.      All figures and table titles need more details to make them self-explanatory. Also, look for supplementary data.

4.      Lines 96-105 need re-writing and if these equations are not developed by authors, then citing the work will make it much easier to explain for readers.

5.      Lines 110 and 112 describe the same thing as LnVar and Var.

6.      Line 125 needs correction for total HYSk as 3 into 3 will be 9.

7.      References 16-18 are incomplete and need either Doi or URL for validation. 

Comments on the Quality of English Language

Some sections and sentences need rewriting and restructuring as explained in comments earlier, which will cover most of the things. It would be great if authors could get it reviewed by some peers as some sections as a whole need improvement. 

Author Response

Dear Reviewer,

Thank you for reviewing our manuscript and providing corrections and comments that will help improve its quality. We have accepted almost all suggested corrections, and the text has been revised accordingly.

In this study, the authors evaluated the resilience indicators in Holstein cows, and they used the multi-trait linear model equation to assess and correlate the indicators. I really appreciate the author’s efforts towards parameter selection to improve the adaptability of dairy cattle. Some sections are well-written and explained well, but I have some suggestions that are critical for the overall improvement of this manuscript.

The abstract section needs improvement in terms of explaining the hypothesis and objectives of the study, so readers can understand the significance of the study. The material and methods (M&M) and discussion sections also need major revisions. In terms of M&M, some equations like 1-4 are explained but I think they are already developed ones, especially equations 1-2, so if authors can reference them effectively then I do not think there is a need to explain them here. Equation 3, explaining the model should be included as explained but other equations if taken from previous studies then citing them will explain everything. The discussion section needs rewriting, although headings are good, there are too many headings for one section, and it should start generally and then move towards headings and limitations as well. The result sections are well written however, all tables and figure legends are not self-explanatory and need more details for the reader to understand them. Addressing these comments will contribute to the overall improvement of the manuscript. I also have some specific comments as mentioned below.

Specific comments:

  1. Question:    Lines 16-17: Improve the overall structure of the sentence.

Answer: The sentence corrected was “The worse resilience in high-producing cows occurred, but on the contrary, more resilient cows showed higher fat and protein contents. “

The sentence was replaced by “Better resilience was associated with lower milk yield but higher milk protein and milk fat contents.” (L16-17, green).

  1. Question:   Lines 55, 69, and 81 need proper citations.

Citations were added to L55 [4] and L81 (L86) [14]. As for the L69, we cited the relevant studies in the same paragraph (L72-L82)

  1. Question:   All figures and table titles need more details to make them self-explanatory. Also, look for supplementary data.

Titles were corrected and description added on L98, L159, L206-209, L212-216, L235-239, L248-252, L255-L259, green).

  1. Question:   Lines 96-105 need re-writing and if these equations are not developed by authors, then citing the work will make it much easier to explain for readers.

Answer: We cited the equations (1, 2) to give a mathematical representation of the procedures used. Sources of both equations are listed in (15,16, L102)

  1. Question: Lines 110 and 112 describe the same thing as LnVar and Var.

Answer: The sentence was re-formulated for better clarity to: The resilience indicators were calculated as the log-transformed variance of DMY (Var), log-transformed variance (LnVar), lag-1 autocorrelation (r-auto) and skewness (skew) of the daily deviations from the fitted individual lactation curves. (L113-116, green)

  1. Question:   Line 125 needs correction for total HYSk as 3 into 3 will be 9.

Answer: Total HYS was not stated, it combined 10 herds with 3 seasons and 3 years. Total number was added (L129-130, green).

  1. References 16-18 are incomplete and need either Doi or URL for validation. 

Answer: The references were corrected (L437-441, green).

Comments on the Quality of English Language

Question: Some sections and sentences need rewriting and restructuring as explained in comments earlier, which will cover most of the things. It would be great if authors could get it reviewed by some peers as some sections as a whole need improvement. 

Answer: The manuscript was edited by Elsevier Language Editing Services before sending to Animals Editorial Office, as showed enclosed Certificate.

Round 2

Reviewer 2 Report

Comments and Suggestions for Authors

Regarding the SE of the estimates, see Meyer, Karin, and David Houle. “Sampling based approximation of confidence intervals for functions of genetic covariance matrices.” Proc. Assoc. Advmt. Anim. Breed. Genet. Vol. 20. 2013. It is not true that REMLF90 does not produce standard errors. An alternative to the frequentist approach used in this study would be to employ a Bayesian approach and estimate the SE from the posterior distribution of the estimates.

Also, the fact that the estimates of the genetic correlations that rely on prior estimates of the EBV for many traits makes these estimates essentially worthless. They should be omitted from the manuscript. Alternatively the authors should obtain the data upon which the EBV were calculated and estimate the genetic correlations properly.

Author Response

Dear reviewer,

We appreciate your review of our manuscript and the corrections and comments that will enhance its quality. We have accepted the suggested changes, and the text has been revised accordingly.

Regarding the SE of the estimates, see Meyer, Karin, and David Houle. “Sampling based approximation of confidence intervals for functions of genetic covariance matrices.” Proc. Assoc. Advmt. Anim. Breed. Genet. Vol. 20. 2013. It is not true that REMLF90 does not produce standard errors. An alternative to the frequentist approach used in this study would be to employ a Bayesian approach and estimate the SE from the posterior distribution of the estimates

A: L173-L178 Table 3. The SD-posterior standard deviations by GIBBSF90 were added (violet).

Also, the fact that the estimates of the genetic correlations that rely on prior estimates of the EBV for many traits makes these estimates essentially worthless. They should be omitted from the manuscript. Alternatively the authors should obtain the data upon which the EBV were calculated and estimate the genetic correlations properly.

A: L144-146: added information on the source of GEBVs/RBVs for traits other than resilience. Correlations were re-calculated, and only RBVs for resilience indicators with reliability >0.35 were used (as in Chen et al., 2023). The results were corrected accordingly (Figures3-6, Results: L223-228, L243-248, L262-265, Discussion: L330-332, L341-344, References 22, 24) (violet)

Reviewer 3 Report

Comments and Suggestions for Authors

I really appreciate the authors' efforts in making it clearer to understand but there are still sections that need further improvement and some of the previous comments have not been addressed yet, which will be highlighted further.

Line 58: There is a need to describe DMY, as it has not been described previously.

Lines 72-75 need correction in terms of structuring since both sentences need to have citations.

Line 86 showing reference 14 is referenced twice so needs correction.

Line 98 showing Table 1 needs updates in terms of units and footnotes.

Lines 100-119 have not been updated yet and still need improvement. If equations are built based on previous literature, then it should be mentioned with citation but since there are no equations in provided studies [16, 17], so I assume there will be justification for building these equations too.

Line 114 shows LnVar, but it does not mention which variance it is.

Line 159, which mentions the title for Table 2, needs improvement in terms of wording. It would be better phrased as 'Descriptive Statistics of ___.' For this reason, it was suggested to have it reviewed by an experienced writer with stronger English fluency.

Lines 180-181 footnotes still need improvement as there is a need to include footnotes in the table as well.

Similarly, all figures later from Figure 2-6 need better sentence structure as well.

Lines 266-363 covering the whole discussion section needed to be started in a better format and should also cover the limitations as mentioned in the conclusion later.

Lines 437-441 need correction for reference 14.

Lines 444-448 still need correction for the reference style of Animals.

Comments on the Quality of English Language

There is a need to better structure discussion sections. Moreover, material and methods needed to be improved as well.

Author Response

Dear reviewer,

We appreciate your review of our manuscript and the corrections and comments that will enhance its quality. We have accepted suggested changes, and the text has been revised accordingly.

Line 58: There is a need to describe DMY, as it has not been described previously.

A: DMY description was added (L57, green)

Lines 72-75 need correction in terms of structuring since both sentences need to have citations.

            A: Citations were added (L72-73)

Line 86 showing reference 14 is referenced twice so needs correction.

            A: Reference was corrected. (L86)

Line 98 showing Table 1 needs updates in terms of units and footnotes.

            A: Table 1 was corrected, units and footnotes added (L98–101, green).

Lines 100-119 have not been updated yet and still need improvement. If equations are built based on previous literature, then it should be mentioned with citation but since there are no equations in provided studies [16, 17], so I assume there will be justification for building these equations too.

            A: Both equations and references were omitted (L102).

Line 114 shows LnVar, but it does not mention which variance it is.

            A: Definition was corrected. (L106-109, green)

Line 159, which mentions the title for Table 2, needs improvement in terms of wording. It would be better phrased as 'Descriptive Statistics of ___.' For this reason, it was suggested to have it reviewed by an experienced writer with stronger English fluency.

A: The title of Table 2 was corrected. (L154)

Lines 180-181 footnotes still need improvement as there is a need to include footnotes in the table as well.

A: Table 3 was corrected (L173-178).

Similarly, all figures later from Figure 2-6 need better sentence structure as well.

A: All titles (Figures2-6) were edited (L202-205, 208-213, green)

Lines 266-363 covering the whole discussion section needed to be started in a better format and should also cover the limitations as mentioned in the conclusion later.

            A: Discussion section was rewritten towards a better structure, with more general start (L267-276), reduced headings (L285, L304), new references added (R22-24) and limitation mentioned (L366-372).

Lines 437-441 need correction for reference 14.

A: corrected

Lines 444-448 still need correction for the reference style of Animals.

A: corrected L445-446 (reference 16 and 17, violet).